

# $^1$H NMR studies distinguish the water soluble metabolomic profiles of untransformed and RAS-transformed cells

Vered Marks[1], Anisleidys Munoz[1,2], Priyamvada Rai[2,3] and Jamie D. Walls[1]

[1] Department of Chemistry, University of Miami Coral Gables, FL, USA
[2] Department of Medicine, Division of Hematology and Oncology, University of Miami Miller School of Medicine, Miami, FL, USA
[3] Sylvester Comprehensive Cancer Center, University of Miami Miller School of Medicine, Miami, FL, USA

## ABSTRACT

Metabolomic profiling is an increasingly important method for identifying potential biomarkers in cancer cells with a view towards improved diagnosis and treatment. Nuclear magnetic resonance (NMR) provides a potentially noninvasive means to accurately characterize differences in the metabolomic profiles of cells. In this work, we use $^1$H NMR to measure the metabolomic profiles of water soluble metabolites extracted from isogenic control and oncogenic HRAS-, KRAS-, and NRAS-transduced BEAS2B lung epithelial cells to determine the robustness of NMR metabolomic profiling in detecting differences between the transformed cells and their untransformed counterparts as well as differences among the RAS-transformed cells. Unique metabolomic signatures between control and RAS-transformed cell lines as well as among the three RAS isoform-transformed lines were found by applying principal component analysis to the NMR data. This study provides a proof of principle demonstration that NMR-based metabolomic profiling can robustly distinguish untransformed and RAS-transformed cells as well as cells transformed with different RAS oncogenic isoforms. Thus, our data may potentially provide new diagnostic signatures for RAS-transformed cells.

## INTRODUCTION

It has long been appreciated that the metabolism of normal and malignant cells can significantly differ (*Warburg, 1956*). Measuring the cellular metabolomic profiles can therefore provide a "snapshot" of the degree of oncogenic malignancy in cancer cells (*Griffin & Schockor, 2004*). One particularly important technique that can measure metabolomic profiles is nuclear magnetic resonance (NMR). NMR is a noninvasive method that can provide highly reproducible and quantitative metabolomic information and has been previously used to detect metabolic fingerprints from a variety of oncogenic pathways (*Morvan & Demidem, 2007*; *Southam et al., 2008*).

One of the earliest applications of cellular NMR metabolomics has been to look for biomarkers associated with activation of the RAS oncogene (*Aboagye & Bhujwalla, 1999*;

Corresponding author
Jamie D. Walls, jwalls@miami.edu

*Ronen et al.*, *2001*). The RAS oncogene, which can exist in either of the HRAS, KRAS, or NRAS isoforms, is found mutated in approximately 30% of all human cancers and produces aggressive, treatment resistant tumors (*Fernandez-Medarde & Santos*, *2011*). Of the three major isoforms, KRAS is found to be the most commonly mutated in human tumors. NRAS is also found activated in certain tumor types such as melanoma, whereas HRAS mutations are rarely found in human cancers (*Bos*, *1989*). The three isoforms, which differ in their membrane-targeting domain, were historically considered to be redundant in their function (*Castellano & Santos*, *2011*). However, a number of studies have shown that the three isoforms are functionally different (*Li, Zhu & Guan*, *2004*; *Walsh & Bar-Sagi*, *2001*; *Yan et al.*, *1998*) with tumors sustaining distinct oncogenic versions of RAS showing different progression characteristics (*Parikh, Subrahmanyam & Ren*, *2007*; *Whitwam et al.*, *2007*). Accordingly, high-throughput noninvasive means of detecting RAS signatures from tumor cells are likely to aid in effective diagnosis and design of treatment regimens that target RAS-specific pathways (*Downward*, *2003*; *Omerovic et al.*, *2008*). However to our knowledge, differences in the metabolomic profiles between normal cells and cells transformed with either of oncogenic HRAS, KRAS, or NRAS have not been previously investigated. Anticipating that there will be robust differences between untransformed and RAS-transformed cells, we employed a cell culture system to validate the NMR methodology described in this study.

Specifically, we analyzed BEAS2B immortalized lung epithelial cells stably transformed with either an empty retroviral vector or with either one of the activated versions of the RAS isoforms, HRAS, KRAS and NRAS, as a proof-of-principle system to determine whether $^1$H NMR-based metabolomics could be used to identify unique metabolomic signatures between the RAS-transformed and control cells as well as among the different RAS isoform-transformed cell lines. The advantage of this cell culture system is the isogenic background among the four cell lines as well as the ability to generate the requisite numbers of stably transformed cells for consistent NMR characterization. Our NMR characterization of the metabolomic profiles indicated that each RAS isoform possesses a distinct metabolomic signature that has bearing on its observed cell-physiologic transformative effects.

## MATERIALS AND METHODS

### DNA constructs and viral transduction

The retroviral pBABE KRASV12, HRASV12 and NRASQ61 DNA constructs were obtained from Addgene. Stable transduction of the pBABE empty vector and the RAS constructs into BEAS2B cells was performed as previously described (*Patel et al.*, *2012*). Transduced cells were selected in 2.5 μg/ml puromycin-containing complete culture media for a minimum period of 5–7 days (corresponding to the time taken for untransduced BEAS2B cells to die completely in selection media). Oncoprotein overexpression relative to the control cells was verified via Western blotting as previously described (*Patel et al.*, *2012*).

### Cell culture

BEAS2B cells were obtained from the American Type Culture Collection. All cells were grown at 37 °C in 21% oxygen and 5% $CO_2$. BEAS2B cells and their derivative lines were

maintained in DMEM:F12 complete base media supplemented with 10% fetal bovine serum and 100 units/ml penicillin-streptomycin. All cell culture reagents were obtained from Life Technologies. For each cell line, ten biological replicates were generated by initially seeding ten different 15 cm dishes (Nunclon) with an initial plating of approximately $1 \times 10^6$ cells for the control cells, $1.5 \times 10^6$ cells for the HRAS- and KRAS-transformed cells, and $2 \times 10^6$ for the NRAS-transformed cells (differences in the initial seedings were used to compensate for differences in cellular growth rates so that by the end of the growth period, approximately the same number of cells for each cell line was obtained). In total, all forty plates were seeded at approximately the same time and were allowed to proliferate for a period of four days with the media changed every 48 h. After four days, the cells were trypsinized for approximately two minutes and counted using a Moxi automatic cell counter (VWR) with size parameters adjusted to exclude apoptotic cells. The average final cell counts were $(1.107 \pm 0.050) \times 10^7$ cells per control sample, and $(1.558 \pm 0.291) \times 10^7$, $(1.486 \pm 0.124) \times 10^7$, and $(1.613 \pm 0.156) \times 10^7$ cells per HRAS-, KRAS-, and NRAS-transformed sample, respectively. This corresponded to an average population doubling time of 27.68 h for the control cells and 28.43 h, 29.02 h, and 31.89 h for the HRAS-, KRAS-, and NRAS-transformed cells, respectively. After counting, the cells were pelleted at 1,500 rpm for 5 min at 4 °C with the pellets immediately snap-frozen in liquid nitrogen and stored at −80 °C.

## Metabolite extraction

The extraction of hydrophilic metabolites from cell pellets was performed using previously established procedures (*Gottschalk et al.*, *2008*). Briefly, cell pellets in a 1.5 ml Eppendorf tubes were resuspended by adding 500 µl of a 2:1 ($v/v$) ice-cold solution of methanol (Sigma-Aldrich) and chloroform (Sigma-Aldrich) followed by 3–5 min of vortexing and manual mixing for at least 10 min until a clear solution was obtained. Next, 250 µl of ice-cold chloroform and 250 µl of ice-cold water were each added to the sample, which was then vortexed for 5–7 min to yield a cloudy solution. The sample was sonicated at room temperature for ten minutes followed by centrifugation at 13,000 rpm for 5 min at 4 °C in order to yield three layers. The hydrophilic layer was transferred to a fresh Eppendorf tube followed by bubbling with nitrogen gas (Airgas) to remove any residual methanol. The samples were placed under a high speed vacuum concentrator at room temperature until dried, and the dried hydrophilic layer was stored in a −80 °C freezer until needed.

## NMR sample preparation, acquisition, and processing

The dried hydrophilic layer was resuspended in 400 µl of deuterated PBS at pH = 7.6 that was prepared as previously reported (*Sambrook, Fritsch & Maniatis*, *1989*). The pH of each sample was adjusted to 7.6 by the addition of either dilute HCl or NaOH as needed to ensure that each metabolite appeared at the same chemical shift in all samples. In each sample, 0.5 µl of a 0.1 M aqueous solution of DSS (Sigma Aldrich) was added for chemical shift referencing. After vortexing, each sample was transferred into a 5 mm NMR tube.

The [1]H NMR spectra were acquired on a 500 MHz Bruker Avance spectrometer (operating at 500.13 MHz for [1]H observation) equipped with a 5 mm TCI 500S2 H-C/N-D-05 Z cryoprobe head at 298 K. Each sample was tuned and matched, reshimmed, and the
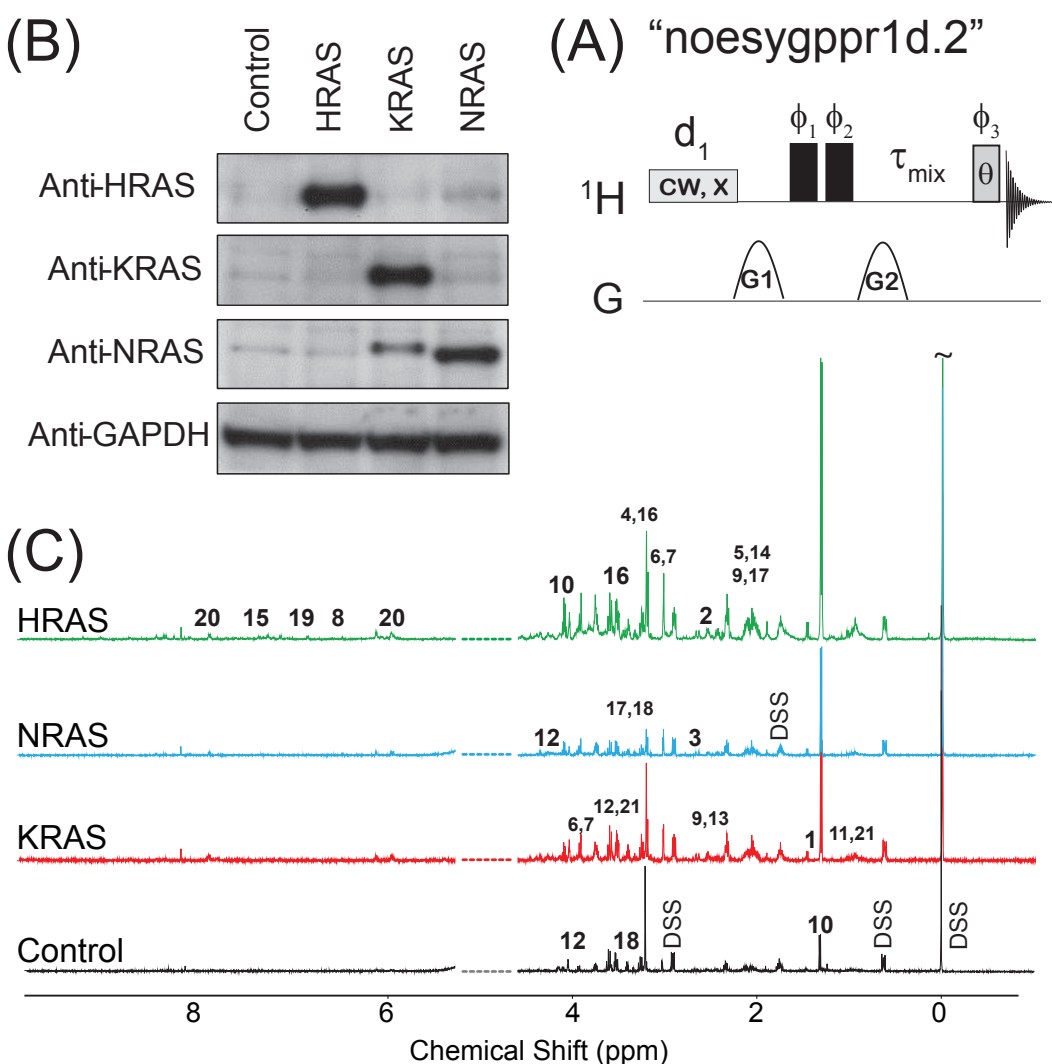

**Figure 1    NOESY pulse sequence, Western Blots, and Representative Spectra.** (A) The 1D NOESY with presaturation pulse sequence. (B) Western blots depicting the control and oncogenic HRAS-, KRAS-, and NRAS-transformed cells. (C) Representative spectra obtained from the 1D NOESY sequence applied to samples made from the control and HRAS-, NRAS-, and KRAS-transformed cells. The spectra were normalized so that the DSS resonance at $\delta = 0$ ppm had the same intensity in all spectra *for display purposes only*. The spectral region for the water resonance is not shown, and certain metabolite resonances are labeled using the codes 1–21 given in Table 1.

90° pulse length was recalibrated (90° pulse lengths ranged between 10 and 12 μs). The $^1$H NMR spectra were acquired using a standard Bruker 1D NOESY pulse program with water presaturation and spoiler gradients applied during the relaxation delay, "noesygppr1d.2" as shown in Fig. 1A. This pulse sequence provides good solvent suppression without rolling baselines (*Beckonert et al., 2007*; *Mckay, 2011*). The following experimental parameters were used in all measurements: sweep width of 10.33 kHz, 65 K acquisition points, a 2 s recycle delay during which a 93 Hz water presaturation pulse was applied, $\tau_{mix} = 101.2$ ms, $\theta = 7.5°–9°$, and 256 scans were acquired for each sample. Half-sine shaped pulsed field

**Table 1  Table of chemical shifts and splitting patterns for metabolites identified by NMR.** List of identified metabolites from the hydrophilic layer, with their corresponding CHEBID, chemical shifts (ppm) and splitting patterns (s, singlet; d, doublet; t, triplet; q, quartet; m, multiplet; dd, doublet of doublets; ddd, doublet of doublet of doublets; bs, broad singlet; bd, broad doublet; bt, broad triplet) used in the Chenomx analysis of the $^1$H spectra. The labels 1–21 are for those metabolites that exhibited a significant difference (adjusted $p$-values $\leq 0.01$) between at least two cell types in either their "effective" NMR metabolite fraction, $x_{metabolite}^{Cell\ type}$ in Eq. (2), or their glutamate normalized signal, $\xi_{\alpha,glutamate}^{Cell\ type}$ in Eq. (3).

| Metabolite [CHEBI ID] | $^1$H chemical shifts (ppm) and multiplicity |
|---|---|
| Acetate [15366] | 1.90(s) |
| Alanine, **1** [16977] | 1.47(d), 3.77(q) |
| Beta-alanine, **2** [16958] | 2.54(t), 3.16(t) |
| Arginine [16467] | 1.64(m), 1.72(m), 1.88(m), 1.92(m), 3.42(t), 3.75(t) |
| AXP [15422, 16027, 16761] | 4.22(m), 4.29(m), 4.39(m), 4.57(t), 4.8(m), 6.14(d), 8.26(s), 8.52(bs) |
| Aspartate, **3** [17053] | 2.67(dd)[a] |
| Choline, **4** [15354] | 3.19(s), 3.51(m), 4.06(m) |
| Choline alfoscerate, **5** [16870] | 2.14(s), 3.22(bs), 3.75(m), 4.54(m) |
| Citrate [30769] | 2.51(d)[a], 2.68(d)[a] |
| Creatine, **6** [16919] | 3.02(s), 3.92(s) |
| Creatine phosphate, **7** [17287] | 3.03(s), 3.94(s) |
| Formate [30751] | 8.44(s) |
| Fumarate, **8** [18012] | 6.51(s) |
| Glutamate [16015] | 2.04(dddd), 2.12(dddd), 2.31(ddd)[a], 2.36(ddd)[a], 3.74(dd) |
| Glutamine, **9** [18050] | 2.10(m), 2.14(m), 2.42(m), 2.47(m), 3.76(t) |
| Glutathione [16856] | 2.14(m), 2.17(m), 2.53(m), 2.57(m), 2.93(dd)[a], 2.97(dd)[a], 3.75(dd)[a], 3.77(dd)[a], 3.79(dd)[a], 4.55(bt) |
| Glycine [15428] | 3.55(s) |
| Isocitrate [151] | 3.02(s), 3.94(s) |
| Isoleucine [17191] | 0.93(t), 0.99(d), 1.25(m), 1.46(m), 1.97(m), 3.66(d) |
| Lactate, **10** [422] | 1.32(d), 4.10(q) |
| Leucine, **11** [15603] | 0.94(d), 0.96(d), 1.67(m), 1.70(m), 1.73(m), 3.70(m) |
| Malate [6650] | 2.35(dd), 2.66(dd), 4.29(bd) |
| Myo-inositol, **12** [17268] | 3.26(t), 3.52(dd), 3.61(dd)[a], 4.05(t) |
| N-acetylaspartate, **13** [21547] | 2.00(s), 2.48(dd), 2.68(dd), 4.38(ddd) |
| N-acetylcysteine, **14** [28939] | 2.07(s), 2.90(dd)[a], 2.93(dd)[a], 4.37(m) |
| N-acetyY[c] [17533] | 1.91(m), 2.03(s), 2.10(m), 2.30(m), 2.33(m), 4.15(m) |
| [NADZ][d] [15846, 16908] | 8.165(s), 8.41(s), 9.33(s) |
| [NADPZ][d] [16474, 18009] | 8.14(s), 8.41(s), 9.29(s) |
| Phenylalanine, **15** [17295] | 3.11(dd), 3.37(dd), 3.98(dd), 7.31(d)[a], 7.36(m), 7.41(m) |
| Phosphocholine, **16** [18132] | 3.21(bs), 3.58(m), 4.15(m) |
| Proline, **17** [17203] | 1.98(m), 2.03(m), 2.06(m), 2.34(m), 3.33(m), 3.41(m), 4.12(dd) |
| Pyruvate [32816] | 2.36(s) |
| Succinate [15741] | 2.39(s) |
| Taurine, **18** [15891] | 3.25(t), 3.41(dd) |

**Table 1** (*continued*)

| Metabolite [CHEBI ID] | $^1$H chemical shifts (ppm) and multiplicity |
|---|---|
| Tyrosine, **19** [17895] | 3.04(dd), 3.18(dd), 3.93(dd), 6.88(d)[a], 7.18(d)[a] |
| UDP-X[b], **20**[17200, 18066, 18307] | 3.44(t), 3.53(td), 3.76(t), 3.78(dd), 3.86(m), 3.89(m), 4.19(ddd)[a], 4.24(ddd)[a], 4.27(m), 4.36(m), 4.37(m), 5.59(dd), 5.96(d), 5.98(bd), 7.94(d) |
| Valine, **21** [16414] | 0.98(d), 1.03(d), 2.26(hd), 3.60(d) |
| DSS | 0.00(s), 0.63(m), 1.76(m), 2.91(m) |

**Notes.**

[a] Multiplet with second-order couplings.

[b] For UDP-X can be UDP-galactose, UDP-glucose, or UDP-glucoranate.

[c] For N-acetylY, the resonances used in the analysis stand for N-acetylglutamate, N-acetylglycine, and/or (and most likely) N-acetylglutamine.

[d] Only the listed resonances were used in the analysis of [NADZ] {[NADH] and/or [NAD$^+$]}, and [NADPZ] {[NADPH] and/or [NADP$^+$]}.

gradients of duration 1 s with maximum gradient strengths of G1 = 24 G/cm and G2 = −23.7 G/cm were used in Fig. 1A along with a 200 μs gradient stabilization delay placed after each gradient pulse. After acquisition, all FIDs were imported into the Chenomx NMR Suite Profiler (version 7.6., Chenomx Inc., Edmonton, Canada). The data were Fourier transformed after multiplication by an exponential window function with a line broadening of 0.5 Hz, and the spectra were manually phase corrected and baseline adjusted using a cubic-spine function. From the initial set of ten biological replicates for each cell line, only 8 of the control, 7 of the HRAS, 9 of the KRAS, and all 10 of the NRAS samples provided measureable NMR signal from resonances other than the solvent peak. Therefore, the results presented in this work represent data obtained from those $N_S = 8$ biological replicates of the control cells, and those $N_S = 7$, $N_S = 9$, and $N_S = 10$ biological replicates of the HRAS-, KRAS-, and NRAS-transformed cells.

The Chenomx NMR Suite Profiler was used to identify metabolites by fitting compound signatures from the provided NMR spectral library. In total, 37 metabolites were identified by NMR. The effective NMR metabolite concentration in each sample, $S_{metabolite}$, was calculated using the Chenomx NMR Suite Profiler by determining the heights of the compound signatures that best fit the sample spectra with the effective concentration of the internal DSS standard being set to $S_{DSS} = 0.1248$ mM, which was the actual DSS concentration in each sample. The table of identified metabolites and their signals was then exported and saved in an Excel worksheet.

## Statistical analysis

The "effective" NMR cellular content for metabolite $\alpha$ (moles/cell) taken from the $s$th biological replicate of a given cell type, $\tilde{C}_{\alpha,s}^{Cell\ type}$, was calculated by multiplying $S_\alpha$ by the NMR sample volume (400.5 μl) and by dividing by the number of cells used to make up each NMR sample. $\tilde{C}_{\alpha,s}^{Cell\ type}$ is related to the *actual* cellular content for metabolite $\alpha$, $C_{\alpha,s}^{Cell\ type}$, by the relationship

$$\tilde{C}_{\alpha,s}^{Cell\ type} = \chi_s^{Cell\ type} f_\alpha C_{\alpha,s}^{Cell\ type} \tag{1}$$

where $\chi_s^{Cell\ type}$ and $f_\alpha$ are dimensionless proportionality factors. The **sample-** and **cell type-independent** factor $f_\alpha$ is taken to depend only upon the experimental NMR

acquisition parameters (such as recycle delays, mixing times, magnetic field strength, etc.) and metabolite $\alpha$'s spin topology and relaxation properties. This contribution can in principle be found by applying the 1D NOESY sequence in Fig. 1A to prepared standards. The **sample-** and **cell type-dependent** factor $\chi_s^{Cell\ type}$ in Eq. (1) is due to the overall metabolite extraction efficiency, which can vary from sample to sample and depends quite sensitively on cell handling (*Duarte et al.*, *2009*) and the particular metabolic quenching and extraction method employed in the study.

The various $\tilde{C}_{\alpha,s}^{Cell\ type}$ were used to calculate the "effective" NMR fraction of metabolite $\alpha$ in each sample, $x_\alpha^{Cell\ type}$, as follows:

$$x_\alpha^{Cell\ type} = \frac{\tilde{C}_\alpha^{Cell\ type}}{\sum_{j=1}^{37} \tilde{C}_j^{Cell\ type}} = \frac{f_\alpha C_\alpha^{Cell\ type}}{\sum_{j=1}^{37} f_j C_j^{Cell\ type}}. \tag{2}$$

As defined in Eq. (2), $x_\alpha^{Cell\ type}$ is dimensionless and independent of the number of cells in a given biological replicate that were used to make the sample. More importantly, $x_\alpha^{Cell\ type}$ is independent of the **sample-dependent** fluctuation factor, $\chi_s^{Cell\ type}$ in Eq. (1). The total intensity normalization in Eq. (2) is analogous to that used in spectral binning analysis commonly employed in NMR metabolomic studies. Furthermore, if the various $f_\alpha$ are identical for each metabolite, i.e., $f_\alpha = f$ for all metabolites, then $x_\alpha^{Cell\ type}$ in Eq. (2) is simply the mole fraction of metabolite $\alpha$ for a given cell type (in general, this is not the case, and $f_\alpha \neq f_\beta$).

An ANOVA test, implemented using the MATLAB function "anova1" available in MATLAB's Statistics toolbox, was first used to test the hypotheses that $\langle x_\alpha^{HRAS} \rangle = \langle x_\alpha^{KRAS} \rangle = \langle x_\alpha^{NRAS} \rangle = \langle x_\alpha^{Control} \rangle$ for each metabolite $\alpha$, where $\langle x^{Cell\ type} \rangle$ represents the average value of $x$ for a given cell type. The BY algorithm (*Benjamini & Yekutieli*, *2001*) implemented in MATLAB (*Groppe*, *2010*) with the false discovery rate set to 0.01 was then applied to the $p$-values from the ANOVA analysis to determine those metabolites where $\langle x_\alpha \rangle$ significantly differed (adjusted $p$-values with $p \leq 0.01$) between at least two cell types. For those metabolites identified by the ANOVA test, further post-hoc/multiple comparison testing using the BY algorithm was performed to identify which pair(s) of cell types $\langle x_\alpha \rangle$ significantly differed (adjusted $p$-values with $p \leq 0.01$, which are given in Table S1). Finally, a PCA of the various $x_\alpha^{Cell\ type}$ was performed using the "pca" command in the Statistics toolbox in MATLAB, which by default, centers the data before performing the PCA.

Due to the similar average values of the "effective" NMR glutamate content, $\langle x_{glutamate} \rangle$, observed in both the control and RAS-transformed cells (Fig. S1) and the relatively large glutamate signals observed in all cells lines (only the lactate signals were larger on average), an alternative to the total intensity normalization scheme used in Eq. (2) was also investigated whereby the metabolite signals were normalized by the observed glutamate signal in each sample. In this case, the glutamate normalized signal for metabolite $\alpha$ is given by:

$$\xi_{\alpha,glutamate}^{Cell\ type} = \frac{\tilde{C}_\alpha^{Cell\ type}}{\tilde{C}_{glutamate}^{Cell\ type}} = \frac{x_\alpha^{Cell\ type}}{x_{glutamate}^{Cell\ type}} = \frac{f_\alpha}{f_{glutamate}} \times \frac{C_\alpha^{Cell\ type}}{C_{glutamate}^{Cell\ type}} \tag{3}$$

$\xi_{\alpha,glutamate}^{Cell\ type}$ is directly proportional to the ratio of the *actual* cellular metabolite $\alpha$ to glutamate content, and, like $x_{\alpha}^{Cell\ type}$ in Eq. (2), it is also independent of the **sample-dependent** fluctuation factor, $\chi_s^{Cell\ type}$. A PCA of the various $\xi_{\alpha,glutamate}^{Cell\ type}$ was also performed, where in this case only 36 metabolites were considered in the analysis since $\xi_{glutamate,glutamate}^{Cell\ type} = 1$ in each sample by definition (Eq. (3)).

One advantage of using glutamate normalization in Eq. (3) compared with using total metabolite normalization in Eq. (2) is that the ratio of $\xi_{\alpha,glutamate}$ between different cell types is independent of the $f_\alpha$ factors and depends only on the actual cellular metabolite contents:

$$\frac{\xi_{\alpha,glutamate}^{Cell\ type\ 1}}{\xi_{\alpha,glutamate}^{Cell\ type\ 2}} = \left(\frac{C_\alpha^{Cell\ type\ 1}}{C_{glutamate}^{Cell\ type\ 1}}\right) \bigg/ \left(\frac{C_\alpha^{Cell\ type\ 2}}{C_{glutamate}^{Cell\ type\ 2}}\right). \tag{4}$$

The ratio in Eq. (4) is equivalent to the relative fold change in the ratio of the *actual* cellular metabolite $\alpha$ to glutamate content between different cell types. In those instances where a significant difference in $\langle\xi_{metabolite,glutamate}\rangle$ between at least two cell types was identified by ANOVA and post-hoc/multiple comparison testing, quantitative confidence intervals for the ratio in Eq. (4) were calculated using Fieller's method for unpaired data (*Motulsky, 1995*). In this case, the $(100-\alpha)\%$ confidence range for $\xi_{metab.,\ glut.}^{Cell\ type\ 1}/\xi_{metab.,\ glut.}^{Cell\ type\ 2}$ in Eq. (4), which is denoted by $\epsilon_{(100-\alpha)\%}\left(\frac{\xi_{metab.,\ glut.}^{Cell\ type\ 1}}{\xi_{metab.,\ glut.}^{Cell\ type\ 2}}\right)$, is given by (*Motulsky, 1995*):

$$
\begin{aligned}
&\epsilon_{(100-\alpha)\%}\left(\frac{\xi_{metab.,\ glut.}^{Cell\ type\ 1}}{\xi_{metab.,\ glut.}^{Cell\ type\ 2}}\right) \\
&= \frac{\mu_1}{(1-g)\mu_2}\left(1 \pm t_{1-\frac{\alpha}{2},N_1+N_2-2}\sqrt{(1-g)\left(\frac{\sigma_1}{\mu_1\sqrt{N_1}}\right)^2 + \left(\frac{\sigma_2}{\mu_2\sqrt{N_2}}\right)^2}\right)
\end{aligned}
\tag{5}
$$

where $N_1$ and $N_2$ are the number of biological replicates of cell types 1 and 2, respectively, $\mu_1$ and $\sigma_1$ are the average and standard deviations for $\xi_{metabolite,glutamate}^{Cell\ type\ 1}$, respectively, $\mu_2$ and $\sigma_2$ are the average and standard deviations for $\xi_{metabolite,glutamate}^{Cell\ type\ 2}$, respectively, $g = \left(\frac{\sigma_2 t_{1-\frac{\alpha}{2},N_1+N_2-2}}{\mu_2\sqrt{N_2}}\right)^2$, and $t_{1-\frac{\alpha}{2},\ N_1+N_2-2}$ is the $\left(1-\frac{\alpha}{2}\right)$th quantile of the $t$-distribution with $N_1+N_2-2$ degrees of freedom. If $g \geq 1$, the relative fold change in cellular metabolite to glutamate content cannot be calculated using Eq. (5). The MATLAB files and commands used in the statistical analysis of the metabolomics data is given as File S1.

## RESULTS AND DISCUSSION

### Generation of isogenic cell lines for the study

The results of immunoblotting total protein lysates from the four cell types against the various RAS isoforms are shown in Fig. 1B. Western blotting with antibodies against HRAS (first lane), KRAS (second lane) and NRAS (third lane) confirmed that the cells expressed the appropriate RAS isoforms. Uniformity of loading was also confirmed by

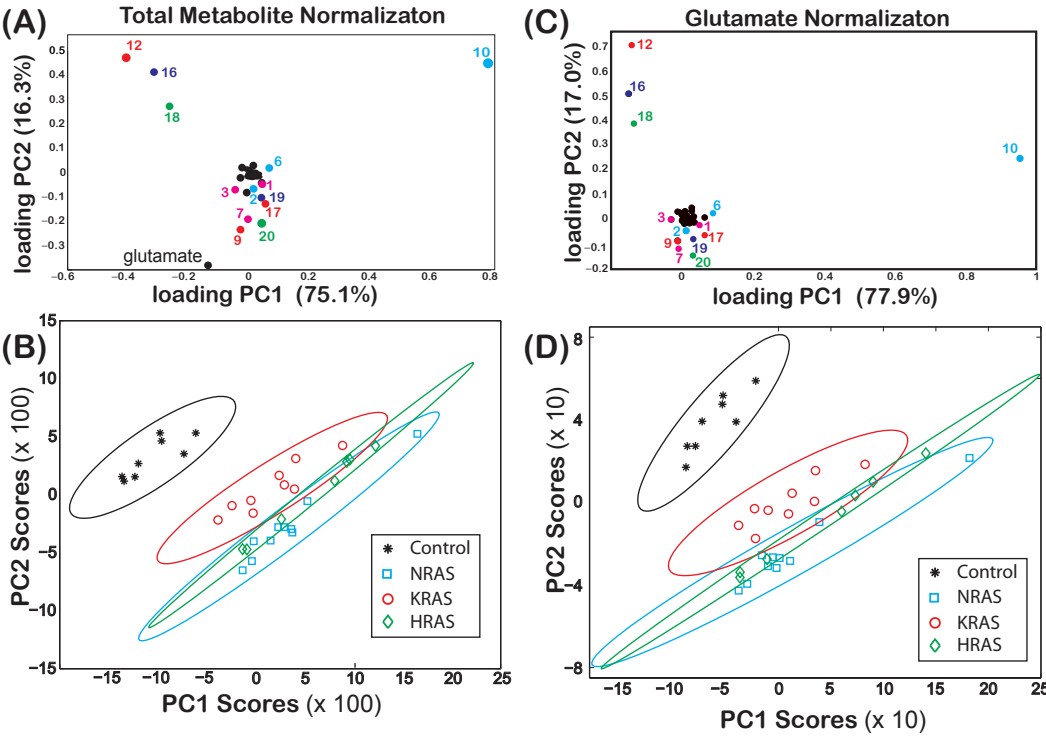

**Figure 2** **Loadings and score plots for effective NMR metabolite fractions.** PCA of the effective NMR metabolite fractions, $x_{metabolite}$ in Eq. (2), and glutamate normalized signals, $\xi_{metabolite,glutamate}$ in Eq. (3), for (asterisks) control and (diamonds) HRAS-, (circles) KRAS-, and (squares) NRAS-transformed cells. Loading plots for $x_{metabolite}$ [(A) PC1 (75.1%) and PC2 (16.3%)] and $\xi_{metabolite,glutamate}$ [(C) PC1 (77.9%) and PC2 (17.0%)] are shown. The identities of certain metabolites are denoted by the labels given in Table 1. Score plots of PC2 versus PC1 of centered data with the corresponding 99% confidence ellipses (*Hoover*, *1984*) are shown for both the (B) $x_{metabolite}$ and (D) $\xi_{metabolite,glutamate}$ data. The results in this figure are from $N_S = 8$ biological replicates of the control cells and $N_S = 7, N_S = 9$, and $N_S = 10$ biological replicates of the HRAS-, KRAS-, and NRAS-transformed cells, respectively.

immunoblotting against GAPDH, a housekeeping gene, as shown in the bottom lane of Fig. 1B. Previous characterizations of these cell lines have also confirmed that the introduction of the RAS oncogene confers soft agar colony growth in these cells, which is indicative of oncogenic transformation (*Rai et al.*, *2011*; *Patel et al.*, *2015*; *Giribaldi et al.*, *2015*).

## NMR-based characterization and PCA of metabolomic profiles

Representative spectra taken from a single biological replicate of the control and HRAS-, KRAS-, and NRAS-transformed cells are shown in Fig. 1C, where the spectra were normalized so that the DSS resonance at $\delta = 0$ ppm had the same intensity in all spectra *for display purposes only*. Certain key metabolites are labeled using the codes, 1–21, given in Table 1.

The loadings of PC1 (score of 75.1%) and PC2 (score of 16.3%) from a PCA of $x_{metabolite}^{Cell\ type}$ are shown in Fig. 2A, where some of the components of both PC1 and PC2 are labeled using the codes given in Table 1. In Fig. 2B, a score plot of PC1 vs. PC2, with the corresponding 99% confidence ellipses (*Hoover*, *1984*) drawn for convenience, shows non-overlapping

grouping between the control and the RAS-transformed cells. Similar results were also observed when performing a PCA of $\xi_{metabolite,glutamate}^{Cell\ type}$ as shown in Figs. 2C and 2D.

Non-overlapping groupings at the 99% confidence level between *all* cell lines were found by plotting $x_{lactate}^{Cell\ type}$ vs. $x_{phosphocholine}^{Cell\ type}$ as shown in Fig. 3A, which were mainly due to differences in the phosphocholine levels between cell types (Fig. 4). Likewise, non-overlapping groupings between *all* cell lines, this time at a slightly lower confidence level of 97.5%, were also found by plotting $\xi_{lactate,glutamate}^{Cell\ type}$ vs. $\xi_{phosphocholine,glutamate}^{Cell\ type}$ as shown in Fig. 3B.

Of the 37 metabolites identified by NMR, an ANOVA analysis indicated that 18 metabolites had a significant (adjusted $p$-values with $p \leq 0.01$) difference in their "effective" NMR metabolite fraction ($x_\alpha$ in Eq. (2)) between at least two of the four cell types. Box plots of $x_{metabolite}^{Cell\ type}$ for these 18 metabolites are given in Fig. 4. Of these 18 metabolites, post-hoc/multiple comparison testing using the BY algorithm found that $x_{metabolite}^{Cell\ type}$ for 17 metabolites was significantly different between the control cells and at least one of the RAS-transformed cell types whereas the "effective" NMR cellular content for 6 metabolites significantly differed in at least two of the three RAS-transformed cell lines (adjusted $p$-values $\leq 0.01$, which are given in Table S1). It should be noted that while the ANOVA analysis indicated that $\langle x_{valine} \rangle$ was unequal between at least two of the four cell types, post-hoc/multiple comparison testing could not identify any significant difference in $\langle x_{valine} \rangle$ between cell lines.

Similarly, an ANOVA analysis of the glutamate normalized metabolite content indicated that 16 metabolites had a significant (adjusted $p$-values with $p \leq 0.01$) difference in $\xi_{metabolite,glutamate}^{Cell\ type}$ (Eq. (3)) between at least two of the four cell types, and box plots of $\xi_{metabolite,glutamate}^{Cell\ type}$ for those 16 metabolites are shown in Fig. 5. Post-hoc/multiple comparison testing indicated that $\xi_{metabolite,glutamate}^{Cell\ type}$ significantly differed between the control cells and at least one of the RAS-transformed cell types for 13 metabolites whereas $\xi_{metabolite,glutamate}^{Cell\ type}$ significantly differed between at least two of the three RAS-transformed cell lines for only 6 metabolites (adjusted $p-$values $\leq 0.01$, which are given in Table S2). It should also be noted that while the ANOVA analysis indicated that $\langle \xi_{CP,glutamate} \rangle$ and $\langle \xi_{glutamine,glutamate} \rangle$ were unequal between at least two of the four cell types, post-hoc/multiple comparison testing could not identify any significant difference in either $\langle \xi_{CP,glutamate} \rangle$ or $\langle \xi_{glutamine,glutamate} \rangle$ between the cell lines. In those instances where a significant difference in $\langle \xi_{metabolite,glutamate} \rangle$ between two cell types was identified by ANOVA and post-hoc/multiple comparison testing, the 99% confidence intervals for the relative fold change in the *actual* cellular metabolite to glutamate content between those cell types, $\epsilon_{99\%} \left( \frac{\xi_{metab.,\ glut.}^{Cell\ type\ 1}}{\xi_{metab.,\ glut.}^{Cell\ type\ 2}} \right)$, were calculated using Fieller's method (Eq. (5)) and are given in Table 2. However, even though significant differences between the RAS-transformed and control cells for $\langle \xi_{fumarate,glutamate} \rangle$ and between the HRAS-transformed and control cells for $\langle \xi_{tyrosine,glutamate} \rangle$ were observed (Fig. 5 and Table S2), the 99% CIs for the relative fold change in the cellular fumarate to glutamate content and the cellular tyrosine to glutamate content could not be calculated due to the small signals and large scatter observed for both tyrosine and fumarate in the control cells (which gave $g > 1$ in Eq. (5)).

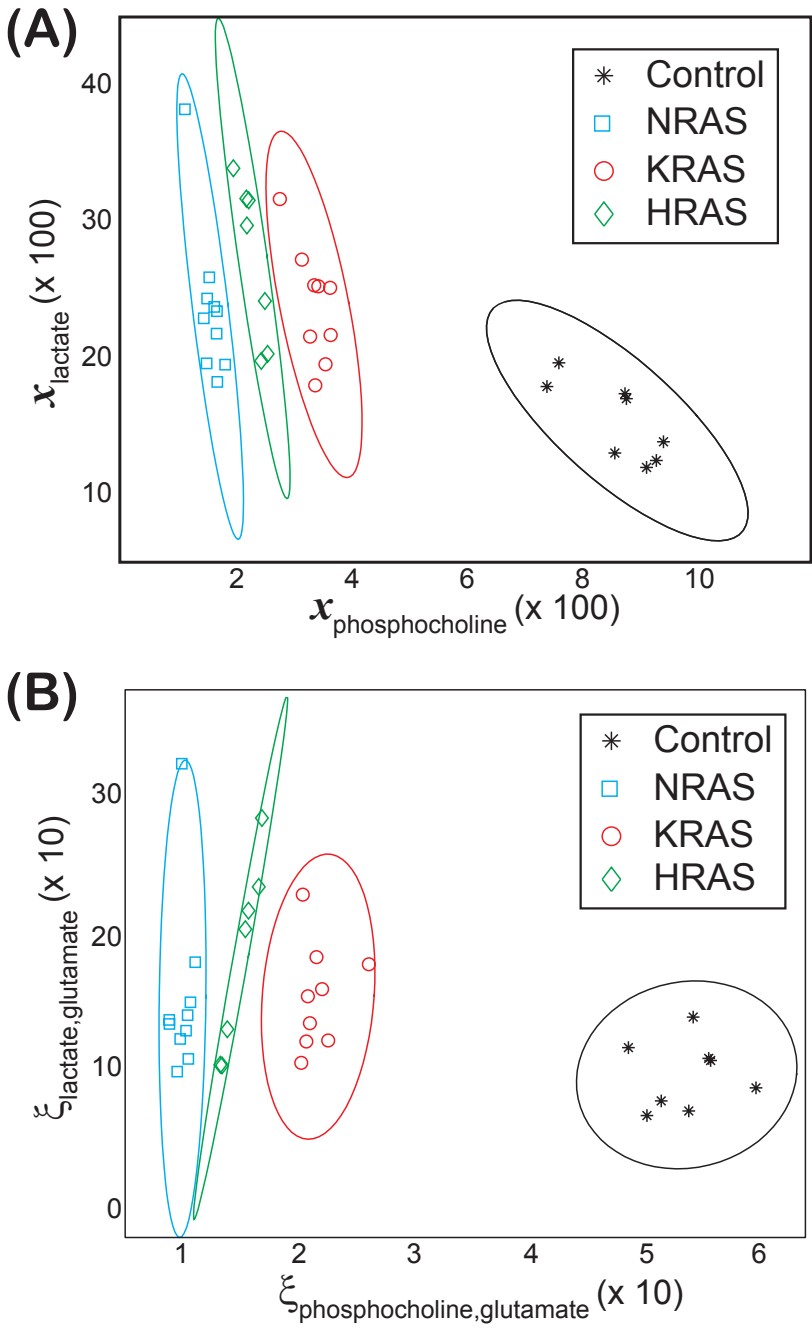

**Figure 3** **Groupings observed for both the lactate vs. phosphocholine NMR metabolite fractions and glutamate normalized signals.** Scatter plots of both (A) the NMR metabolite fractions for lactate, $x_{lactate}^{Cell\ type}$, versus phosphocholine, $x_{phosphocholine}^{Cell\ type}$ and (B) the glutamate normalized lactate, $\xi_{lactate,\ glutamate}^{Cell\ type}$, versus phosphocholine, $\xi_{phosphocholine,glutamate}^{Cell\ type}$, found in the (asterisks) control and (diamonds) HRAS-, (circles) KRAS-, and (squares) NRAS-transformed cells. Confidence ellipses (*Hoover, 1984*) indicate that non-overlapping groupings for all four cell types can be observed at the (A) 99% for the NMR metabolites fractions and at the (B) 97.5% confidence levels for the glutamate normalized signals. In both cases, the results are from $N_S = 8$ biological replicates of the control cells and $N_S = 7$, $N_S = 9$, and $N_S = 10$ biological replicates of the HRAS-, KRAS-, and NRAS-transformed cells, respectively, are shown.

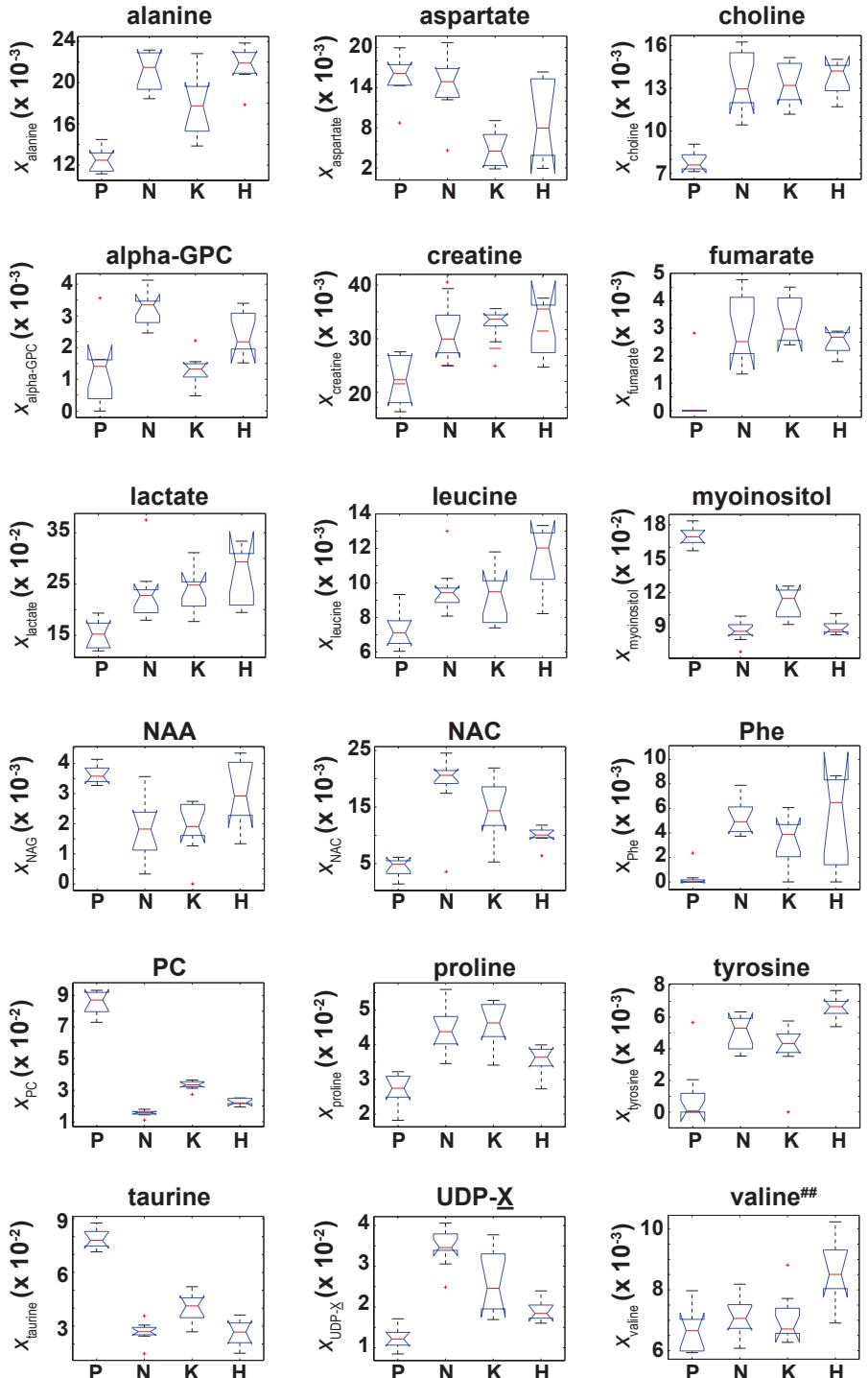

**Figure 4** **Box plots of the NMR metabolite fractions identified by ANOVA analysis.** Box plots of $x_{metabolite}^{Cell\ type}$ for those 18 metabolites identified by an ANOVA analysis which indicated $\langle x_{metabolite} \rangle$ was unequal between at least two of the four cell types. While the ANOVA analysis identified $\langle x_{valine} \rangle$, post-hoc/multiple comparison testing could not identify any significant differences in $\langle x_{valine} \rangle$ between the cell lines, which is denoted by the superscript '##'.

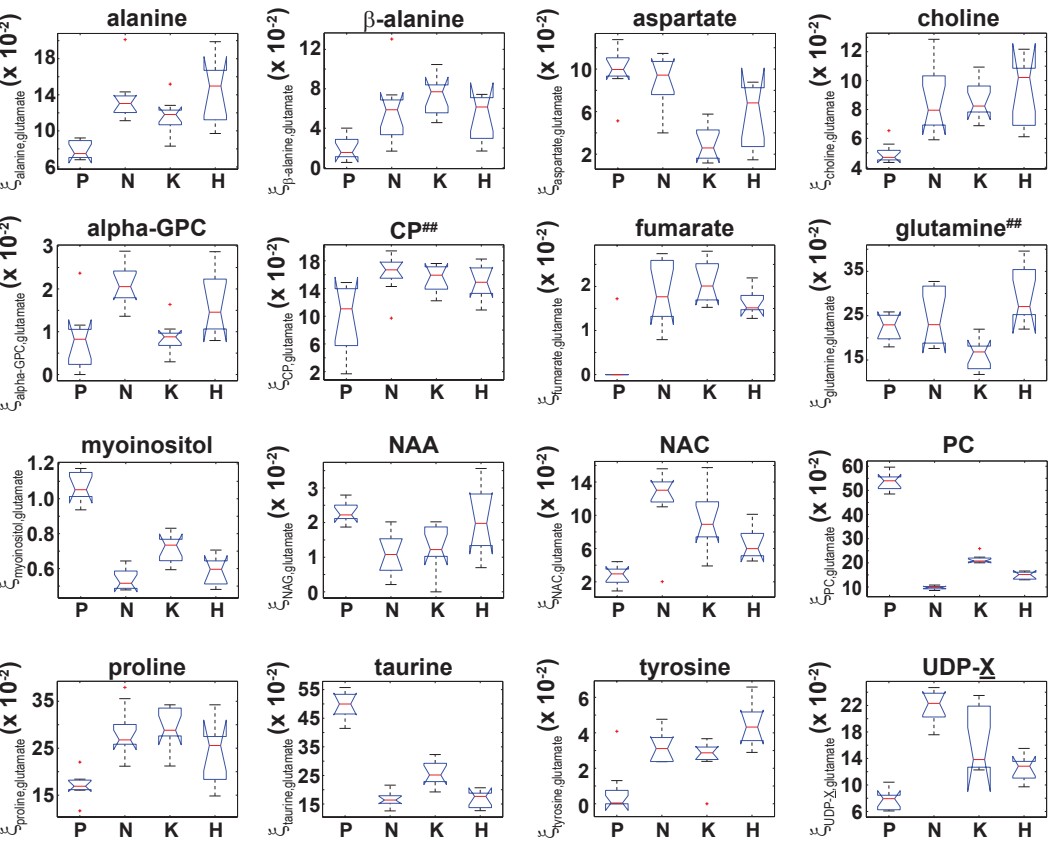

**Figure 5  Box plots of the glutamate normalized signals identified by ANOVA analysis.** Box plots of $\xi_{metabolite,glutamate}^{Cell\ type}$ for those 16 metabolites that were identified by an ANOVA analysis that indicated $\langle \xi_{metabolite,glutamate} \rangle$ was unequal between at least two of the four cell types. While the ANOVA analysis identified both $\langle \xi_{CP,glutamate} \rangle$ and $\langle \xi_{glutamine,glutamate} \rangle$, post-hoc/multiple comparison testing could not identify any significant differences in either $\langle \xi_{CP,glutamate} \rangle$ and $\langle \xi_{glutamine,glutamate} \rangle$ between the cell lines, which is denoted by the superscript '##'.

## NMR-based identification of metabolite differences among the transformed and control cell lines reflect RAS-driven physiologic alterations

Due to the Warburg effect (*Warburg*, *1956*), it is well known that oncogenic transformed cells undergo aerobic glycolysis as opposed to oxidative phosphorylation (*Dang*, *2012*). To establish the validity of our data against known metabolic changes, we assessed how differences in the NMR signals from lactate and alanine, two major byproducts of glycolytic metabolism (*DeBerardinis et al.*, *2007*), varied among the different cell lines. The lactate NMR signal was the largest NMR signal observed in all cell lines (Fig. 4), and lactate was also the largest component to PC1 in the PCA analyses of both $x_{metabolite}$ and $\xi_{metabolite,glutamate}$ in Fig. 2. In fact, $x_{lactate}^{HRAS}$ and $x_{lactate}^{KRAS}$ were found to be statistically larger than $x_{lactate}^{Control}$ (Fig. 4 and Table S1). The cellular alanine to glutamate content was found to be elevated between 50%–75% in KRAS- and NRAS-transformed cells relative to the control cells (Table 2), which is consistent with the reported phenotype of increased aerobic glycolysis in oncogenic

**Table 2** **99% confidence intervals for relative fold change in the ratio of actual cellular metabolite to glutamate content between cell types.** 99% confidence intervals (CIs) for the relative fold change in glutmate normalized signals between cell lines calculated using Fieller's method (*Motulsky, 1995*) in Eq. (5). The lower and upper limits of the 99% CIs are denoted by subscripts that bracket the middle of the CI interval (*Louis & Zeger, 2009*). The abbreviation, n.s., indicates those cases when there was no significant statistical difference in $\langle\xi_{metabolite,glutamate}\rangle$ found between cell lines from post-hoc testing using the BY algorithm (*Benjamini & Yekutieli, 2001*) at a false discovery rate of 0.01. The 99% CIs for the relative fold changes in the cellular fumarate to glutamate content in all RAS-transformed cells relative to control cells and in the cellular tyrosine to glutamate content in HRAS-transformed cells relative to control cells could not be calculated due to the small signals and large scatter of fumarate and tyrosine observed in the control cells (which gave $g > 1$ in Eq. (5)).

| Metabolite $\alpha$ | $\varepsilon_{99\%}\left(\frac{\xi^{HRAS}_{\alpha,glutamate}}{\xi^{Control}_{\alpha,glutamate}}\right)$ | $\varepsilon_{99\%}\left(\frac{\xi^{KRAS}_{\alpha,glutamate}}{\xi^{Control}_{\alpha,glutamate}}\right)$ | $\varepsilon_{99\%}\left(\frac{\xi^{NRAS}_{\alpha,glutamate}}{\xi^{Control}_{\alpha,glutamate}}\right)$ |
|---|---|---|---|
| Alanine | n.s. | $_{1.19}1.5_{1.81}$ | $_{1.37}1.75_{2.13}$ |
| $\beta$-alanine | n.s. | $_{2.07}6.34_{10.61}$ | n.s. |
| Choline | n.s. | $_{1.40}1.79_{2.18}$ | $_{1.28}1.78_{2.28}$ |
| N-acetylcysteine | n.s. | $_{1.86}4.11_{6.37}$ | $_{2.68}5.34_{8.01}$ |
| Proline | n.s. | $_{1.39}1.79_{2.20}$ | $_{1.31}1.72_{2.13}$ |
| UDP-$\underline{X}$ | $_{1.24}1.69_{2.15}$ | $_{1.48}2.25_{3.03}$ | $_{2.28}2.94_{3.60}$ |
| Aspartate | n.s. | $_{0.14}0.33_{0.52}$ | n.s. |
| Myo-inositol | $_{0.45}0.55_{0.65}$ | $_{0.58}0.68_{0.77}$ | $_{0.44}0.51_{0.58}$ |
| N-acetylaspartate | n.s. | n.s. | $_{0.22}0.47_{0.72}$ |
| Phosphocholine | $_{0.24}0.28_{0.31}$ | $_{0.36}0.40_{0.45}$ | $_{0.16}0.18_{0.20}$ |
| Taurine | $_{0.26}0.34_{0.42}$ | $_{0.42}0.52_{0.62}$ | $_{0.28}0.34_{0.40}$ |

| Metabolite $\alpha$ | $\varepsilon_{99\%}\left(\frac{\xi^{NRAS}_{\alpha,glutamate}}{\xi^{HRAS}_{\alpha,glutamate}}\right)$ | $\varepsilon_{99\%}\left(\frac{\xi^{KRAS}_{\alpha,glutamate}}{\xi^{NRAS}_{\alpha,glutamate}}\right)$ | $\varepsilon_{99\%}\left(\frac{\xi^{KRAS}_{\alpha,glutamate}}{\xi^{HRAS}_{\alpha,glutamate}}\right)$ |
|---|---|---|---|
| Aspartate | n.s. | $_{0.15}0.36_{0.56}$ | n.s. |
| Choline alfoscerate | n.s. | $_{0.22}0.44_{0.65}$ | n.s. |
| Myo-inositol | n.s. | $_{1.14}1.34_{1.55}$ | n.s. |
| Phosphocholine | $_{0.58}0.67_{0.76}$ | $_{1.99}2.23_{2.47}$ | $_{1.27}1.48_{1.69}$ |
| Taurine | n.s. | $_{1.23}1.57_{1.91}$ | $_{1.19}1.61_{2.04}$ |
| UDP-$\underline{X}$ | $_{1.44}1.80_{2.16}$ | n.s. | n.s. |

RAS-transformed cells (*Hahn & Weinberg, 2002*), although we should point out that our study provides only a steady-state snapshot of the metabolic profile.

The cellular UDP-$\underline{X}$ (i.e., UDP-glucose, UDP-galactose, and/or UDP-glucourinate), which are important molecules in glucose metabolism and in the formation of cellular polysaccharides (*Berg, Tymoczko & Stryer, 2002*), to glutamate content was elevated between a factor of 1.69–2.94 in the RAS-transformed cells lines relative to the control cells (Table 2). Likewise, the cellular N-acetylcysteine, a thiolic antioxidant (*Oikawa et al., 1999*), to glutamate content was also elevated in all RAS-transformed cells relative to control cells (Fig. 5) with statistically significant differences occurring for the KRAS- and NRAS-transformed cells, where $\frac{C_{NAC}}{C_{glutamate}}$ was 4.11 and 5.34 times larger relative to control cells, respectively (Table 2). The elevated levels of N-acetylcysteine in RAS-transformed cells is a significant finding given that RAS-transformed cells are known to exhibit elevated redox protective mechanisms (*Young et al., 2004*).

The metabolomic signatures of two cellular osmolytes, taurine and myo-inositol, also showed significant differences between the RAS-transformed and control cells. The cellular taurine to glutamate content and the cellular myo-inositol to glutamate content were between 50–66% and 32–45% smaller in all RAS-transformed cells relative to the control cells, respectively (Table 2). As osmolytes regulate the apoptotic cell death pathway (*Lang et al.*, *2005*), the functional relevance of the lower values of $\frac{C_{myo-inositol}}{C_{glutamate}}$ and $\frac{C_{taurine}}{C_{glutamate}}$ observed in RAS-transformed cells may be related to their relative resistance to stress-induced programmed cell death.

An unexpected result from our study was the cellular phosphocholine to glutamate levels. Choline metabolism is an important component in lipid biogenesis (*Glunde, Bhujwalla & Ronen*, *2011*). The cellular phosphocholine to glutamate content in our study was between 60–82% smaller in the RAS-transformed cells relative to the control cells, and statistically significant differences among the RAS-transformed cells were also observed (Fig. 5 and Table S2). Similarly, the cellular choline to glutamate content was around 1.78 times larger in the KRAS- and NRAS-transformed cells relative to the control cells (Table 2). Although there are reports indicating phosphocholine levels correlate with elevated malignancy (*Aboagye & Bhujwalla*, *1999*; *Ronen et al.*, *2001*), exceptions in the published literature suggest that this conclusion may be specific to the RAS isoform and cell type being studied (*Eliyahu, Kreizman & Degani*, *2007*).

We have demonstrated in this work that $^1$H NMR can be used to identify unique metabolomic signatures between BEAS-2B immortalized lung epithelial cells and those transformed with the isoforms of the RAS oncogene as well as among the three RAS isoforms. Collectively, our results suggest that measuring cellular metabolomic profiles can help in distinguishing between normal and RAS-transformed cells along with potentially distinguishing among cancer cells expressing different RAS isoforms. In the future, these results may aid in the development of potential screening technology to determine particular cancer treatment regimens.

## ACKNOWLEDGEMENTS

The authors wish to thank Dr. Danny Yakoub (UM) for comments about the manuscript.

### Funding

JDW received support from the Camille and Henry Dreyfus Foundation and the University of Miami. PR received support from a James and Esther King Florida Biomedical New Investigator Research Grant (09KN-11) and an NIH RO1 grant (RO1 CA175086). Support was also received from the National Science Foundation under CHE—1056846. The funders had no role in study design, data collection and analysis, decision to publish, or preparation of the manuscript.

## Grant Disclosures

The following grant information was disclosed by the authors:
Camille and Henry Dreyfus Foundation.
University of Miami.
James and Esther King Florida Biomedical New Investigator Research Grant: 09KN-11.
National Institutes of Health: R01 CA175086.
National Science Foundation: CHE—1056846.

## Competing Interests

The authors declare there are no competing interests.

## Author Contributions

- Vered Marks and Anisleidys Munoz performed the experiments, wrote the paper, prepared figures and/or tables, reviewed drafts of the paper.
- Priyamvada Rai conceived and designed the experiments, contributed reagents/materials/analysis tools, wrote the paper, prepared figures and/or tables, reviewed drafts of the paper.
- Jamie D. Walls conceived and designed the experiments, analyzed the data, contributed reagents/materials/analysis tools, wrote the paper, prepared figures and/or tables, reviewed drafts of the paper.

## Data Availability

 The raw data has been supplied as Data S1.

## Supplemental Information

Supplemental information for this article can be found online at http://dx.doi.org/10.7717/peerj.2104#supplemental-information.

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
