# Peer review of "H NMR studies distinguish the water soluble metabolomic profiles of untransformed and RAS-transformed cells"

_PeerJ, doi:10.7717/peerj.2104_

## Round 0.1 · original submission · Major Revisions

Dear Jamie,

Thank you for submitting your manuscript for publication in PeerJ. It has been examined by two expert reviewers. While reviewer 2 is generally positive about your manuscript, reviewer 2 suggested to reject the manuscript in its present form, as he doubts your experimental design and questions your statement that "the various fmetabolite are sample-independent" and thus doubting the validity of your ANOVA analysis. I have to take these doubts very serious and I agree with the reviewer that in a re-submission you should base the ANOVA on the x values ("effective" NMR fractions of metabolite alpha in each sample as defined in Eq. 1) to get truly independent of sampling and extraction variations. If you consider a re-submission, please respond in a rebuttal letter to every individual point raised by reviewer 1 and take also into account the remark of reviewer 2 regarding different values of the mixing time in 1D NOESY that might affect the intensity of single peaks in the NMR.

I am sorry that I cannot be more positive this time and I hope that you consider the reviews useful in improving the manuscript.

Reviewer 1 ·

Basic reporting

I have no way of checking whether the submission adheres to all PeerJ policies.

Otherwise no comments.

Experimental design

Failed.

Due to the major problem described below, I believe that the whole statistical analysis was done on a doubtful basis of data and therefore cannot be used to make any conclusions:

Major problem:

ch2.4: p.9, "Assuming that the various fmetabolite are sample-independent..."
You cannot make this assumption. fmetabolite is governed by sampling, extraction, all the manipulations that you're doing with your samples. Once, you might be lucky and extract 80% of your metabolite content, next time it is only 75%. this will affect all metabolites equally, but it will not be the same in all samples. We have done similar experiments and seen a large variation between samples taken from the same cell culture. The authors write that some samples did not contain any observable signals - another indication that the sampling procedure has varying efficiency.
If your assumption about fmetabolite being sample-independent fails, it means that you canNOT compare effective NMR cellular contents c~ between samples - and not the absolute cellular concentrations c, either. Unfortunately, your whole statistical analysis is based on those values. Since the values depend on the entirely unknown efficiency of your sampling process, any results from the statistics become doubtful. A higher extraction efficiency on one type of cells (either by chance or because of different properties of this cell-line) will suddenly be interpreted as a higher concentration of the metabolite in this type of cell.
Looking at the Excel file with the c~ of all metabolites, it is seen that the sum of all c~ for particular samples varies widely between 0.29 (sample N6w) and 1.65 (K9w). c~ within the NRAS group varies from 0.29 - 0.94. Why should this be the case if fmetabolite was sample-independent? This would actually mean that some cells have a very diluted cytosol compared to others - this does not make sense.
The average of the sums of c~ is actually significantly lower for the control (0.63) than for the transformed cells - a finding later misinterpreted (line 243.. It should be noted that while the “effective” NMR cellular phenylalanine, tyrosine, and fumarate contents were all found to be significantly larger in the RAS-transformed cells compared to the control cells." and many more occurrences of metabolites that show increased values in RAS-tranformed cells)


Minor points:

ch2.4: last sentence on p.8 plus first on p.9 "The other contribution..."
I disagree. the whole point of the chenomX software is actually to annihilate exactly that part of the contribution. that, of course, requires that the timings of the NOESY pulse sequence is kept strictly to ChenomX requirements. Then, ChenomX would actually account for differences in T1 and its relation to the recycling delays, and for the influence of the mixing times.
Why would the field strength have any influence at all?
I further disagree with the following sentence "This contribution can in principle...": in order for this to work, you would have to take a standard with exactly the same concentration as you have in your cells - but that is the unknown value, that you're trying to find.


ch. 2.3.: ... an ice-cold solution of chloroform and water..": since the two don't mix, please rephrase, I would understand this as "250uL of chloroform and 250uL of water..."

ch 2.4.: why did you use a 7.5-9 degree pulse as read-out pulse after the NOESY mixing time? When was it 7.5 or 9 or anything in between? How was the exact angle determined? Why not 90deg? You are "throwing away" a large part of your signal?
Your timing is not exactly according to the ChenomX recommendations.

ch2.4; 400.5uL is in my experience not enough volume to obtain a good field homogeneity in a 5mm probe using 5mm tubes. Was there anything special about your probe or your tubes?

ch2.4: were metabolites "identified" by help of ChenomX only? Was there any verification by 2D-NMR techniques such as TOCSY or HSQC? Especially singe-peak metabolites can easily be misassigned without HSQC information. In this work, this would be glycine, acetate, malate, succinate and pyruvate.

ch2.4: did you really use 65K acquisition points? 64K =65536 would be more common?

Validity of the findings

The PCA findings seem to be valid and interesting.

However, the ANOVA analysis was calculated on flawed data and is as such not useful at all. Unfortunately, a large part of the discussion is based on this analysis.

Additional comments

You might be able to "rescue" the work by re-doing the ANOVA on the x values ("effective" NMR fractions of metabolite alpha in each sample as defined in Eq. 1). They are truly independent of all sampling and extraction variations (at least if falpha/f is constant over all samples). It corresponds to a "total intensity normalization", a method with some drawbacks, but nevertheless generally accepted. PCA does indicate significant differences between sample classes.

Also, please consider representing your data as "box plots" instead of averages (Table 1).

·

Basic reporting

The authors present a metabolomic study on untransformed and RAS-transformed cells by using data from 1H NMR spectroscopy which are heavily processed by statistical analysis. The aim behind this study is to find out biomarkers associated with the activation of oncogenic RAS gene which was reported to be mutated in many human cancers. The work consists of three different parts, i.e. the preparation of metabolite extractions from RAS transformed cells, NMR experiments and data processing by statistical methods.
The manuscript is very well written without typographical errors. The description of all methods is clear and well understandable. All necessary details can be found either in the manuscript or in Supporting Information.

Experimental design

I know that 1D-NOESY sequence is a standard NMR method for acquiring data for metabolomic profiles due to excellent water suppression. The method is based on different relaxation properties of solutes and water. The mixing time was set up to 101 ms and I assume that the value was taken from Chenomx prescription. However, because the main aim of the study is to prove that the methodology is capable to discriminate metabolic profiles of untransformed and RAS-transformed cells, I would expect that trying at least several different values of the mixing time in 1D NOESY might affect the intensity of single peaks in the NMR spectra and maybe the ratios of the metabolites.

Validity of the findings

The authors managed to find out enough metabolites which differed both from the blind sample and also even among the three isoforms of RAS oncogene mutually. The conclusions are supported by results which are clearly described in the paper without any uncorroborated speculations.

Additional comments

I have just one comment which is described in the Experiemntal Design.

---

## Round 0.2 · accepted · Accept

Both reviewers were very satisfied with the extensive revision you did, and the manuscript comes out nice now.

During production or proof check, please keep an eye on the minor issue in Lines 133, 134, so that in the final version the sentence reads correctly "90 deg pulse" instead of "90 pulse" as it is now.

You also might also consider to add one explanatory sentence to
Line 211-213: "One advantage of using glutamate normalization [..] can be seen by calculating the ratios [..]", where one of the reviewers misses an explanation of the advantage.

Reviewer 1 ·

Basic reporting

The authors have improved legibility and streamlined the report since the first submission.

Minor issue:

Line 211-213: "One advantage of using glutamate normalization [..] can be seen by calculating the ratios [..]"
I cannot see the advantage, and it is not explained, either.

Experimental design

OK.

Validity of the findings

The authors have normaled their concentration data to total intensity and to glutamate concentrations. Thereby, my issues with the first submission have been solved.

·

Basic reporting

No comments.

Experimental design

No comments.

Validity of the findings

No comments.

Additional comments

I have read the revised manuscript and found only one minor typographical error.
Lines 133, 134: 90 pulse should be corrected to 90 deg pulse
Otherwise, I am satisfied with the author’s response to my comment.